# A CRISPR-Cas9 delivery system for in vivo screening of genes in the immune system

Martin W. LaFleur [1,2,3], Thao H. Nguyen[1,3], Matthew A. Coxe[1,3], Kathleen B. Yates [2,4], Justin D. Trombley [1,3], Sarah A. Weiss [2], Flavian D. Brown[1,2,3], Jacob E. Gillis[1,3], Daniel J. Coxe[5], John G. Doench [4], W. Nicholas Haining [2,4] & Arlene H. Sharpe[1,3,4]

Therapies that target the function of immune cells have significant clinical efficacy in diseases such as cancer and autoimmunity. Although functional genomics has accelerated therapeutic target discovery in cancer, its use in primary immune cells is limited because vector delivery is inefficient and can perturb cell states. Here we describe CHIME: CHimeric IMmune Editing, a CRISPR-Cas9 bone marrow delivery system to rapidly evaluate gene function in innate and adaptive immune cells in vivo without ex vivo manipulation of these mature lineages. This approach enables efficient deletion of genes of interest in major immune lineages without altering their development or function. We use this approach to perform an in vivo pooled genetic screen and identify Ptpn2 as a negative regulator of CD8$^+$ T cell-mediated responses to LCMV Clone 13 viral infection. These findings indicate that this genetic platform can enable rapid target discovery through pooled screening in immune cells in vivo.

[1] Department of Microbiology and Immunobiology, Harvard Medical School, Boston, MA 02115, USA. [2] Department of Pediatric Oncology, Dana-Farber Cancer Institute, Boston, MA 02115, USA. [3] Evergrande Center for Immunological Diseases, Harvard Medical School and Brigham and Women's Hospital, Boston, MA 02115, USA. [4] Broad Institute of Harvard and Massachusetts Institute of Technology, Cambridge, MA 02142, USA. [5] School of Energy, Matter, and Transport Engineering at Arizona State University, Tempe, AZ 85287, USA. These authors jointly supervised this work: W. Nicholas Haining, Arlene H. Sharpe. Correspondence and requests for materials should be addressed to W.N.H. (email: Nicholas_Haining@dfci.harvard.edu) or to A.H.S. (email: Arlene_Sharpe@hms.harvard.edu)

Understanding the mechanisms that regulate innate and adaptive immunity has accelerated the development of immunotherapies for autoimmune and allergic diseases, transplant rejection and cancer[1,2]. The dramatic clinical success of immune checkpoint blockade in a broad range of cancers illustrates how fundamental knowledge of immunoregulation can translate to therapy[3]. However, limitations in the tools available for perturbing genes of interest in immune populations has hindered the discovery and validation of new therapeutic targets for immune-mediated diseases.

The use of functional genomics and genetic perturbation strategies has provided an effective tool for the rapid discovery of new therapeutic targets in cancer[4]. In particular, shRNA-based screening enabled the classification of tumor suppressors and essential genes in cancer[5,6]. However, shRNA approaches are limited by the issues of incomplete knockdown and a high degree of off-target effects[7]. Targeted nucleases, such as TALENs and zinc finger nucleases, have enabled the complete knockout of gene targets with improved specificity but require custom design of proteins for each target gene[8,9], making screening difficult. CRISPR-Cas9 genome editing methods to knockout genes in mammalian cells have the advantages of targeted nuclease editing with improved modularity[10–12]. Furthermore, CRISPR-Cas9 screening provides several advantages over shRNA-based approaches, such as improved consistency across distinct sgRNAs and higher validation rates for scoring genes[13].

Genetic perturbation approaches in immune cells have the potential to accelerate the discovery and validation of new therapeutic targets[14]. One current approach is to stimulate T cells to allow transduction with a shRNA/sgRNA-expressing lentiviral vector[15–18] followed by in vitro analysis or in vivo transfer of edited T cells. Although this method is rapid, in vitro stimulation of T cells perturbs their long-term differentiation[19], does not allow for the study of genes expressed during T cell priming, and is only applicable to immune cell populations that are easily transferred intravenously for analysis in disease models. To circumvent some of these issues, we have previously used a system of lentiviral transduction of bone marrow precursors and subsequent creation of bone marrow chimeras for shRNA-based perturbation of naive T cells without disrupting their differentiation or homeostasis[19]. CRISPR-Cas9 transduction of bone marrow precursors has enabled editing of genes involved in oncogenesis to model hematologic malignancies[20–22] and in the development of hematopoietic precursors[23]. However, these approaches have not been used for studying the immune response in different disease models or discovery of regulators of T cell responses during cancer and viral infection.

Here we describe CHIME, a bone marrow chimera-based Cas9-sgRNA delivery system that enables rapid in vivo deletion of immunologic genes of interest without altering the differentiation of mature immune cells. We demonstrate the versatility of this system to delete genes of interest in all major immune cell lineages. As a proof of concept, we perform a curated in vivo screen in the LCMV Clone 13 infection model and show that deletion of *Ptpn2* enhances CD8[+] T cell responses to LCMV Clone 13, thereby revealing a negative regulatory role for *Ptpn2* in CD8[+] T cell-mediated responses to LCMV Clone 13. Our results illustrate the ability of this genetic platform to enable rapid discovery of therapeutic targets in immune cells using pooled loss-of-function screening.

## Results

**CHIME enables efficient deletion of immunologic genes**. To create gene deletions in hematopoietic lineages, we developed a single guide RNA (sgRNA) chimera delivery system using bone marrow from Cas9-expressing mice[24] (Fig. 1a). We isolated Cas9-expressing Lineage[−] Sca-1[+] c-Kit[+] (LSK) cells from donor mice (Supplementary Fig. 1a), transduced the LSK cells with a lentiviral sgRNA expression vector containing a Vex (violet-excited GFP) fluorescent reporter, and transferred the LSK cells to irradiated recipients to create bone marrow chimeric mice. Following 8 weeks of immune reconstitution we isolated immune cells that express Cas9 and the sgRNA (marked by Vex). To determine whether candidate genes from major immune lineages could be deleted in vivo we designed sgRNAs to canonical genes expressed by B cells (*Ms4a1*), macrophages (*Fcgr1*), and dendritic cells (*Ly75*), and created chimeric mice using either these sgRNAs or control sgRNAs. We found that CD20 was significantly reduced on B cells in the presence of two *Ms4a1* (*Cd20*) sgRNAs but not a control sgRNA, demonstrating that in vivo deletion of genes in B cells was possible (Fig. 1b, c and Supplementary Fig. 1b). We next confirmed that *Fcgr1* (*Cd64*) could be deleted in red-pulp splenic macrophages by showing that CD64 expression was significantly reduced for two *Fcgr1* targeting sgRNAs but not a control sgRNA (Fig. 1b, c and Supplementary Fig. 1c). Lastly, we created bone marrow chimeras with two *Ly75* (*Dec205*) targeting sgRNAs or a control sgRNA and showed a significant reduction in DEC205 expression on splenic cross-presenting dendritic cells with *Ly75* targeting sgRNAs but not control sgRNAs (Fig. 1b, c and Supplementary Fig. 1d). Thus, this chimeric system can be used to efficiently delete genes of interest in both innate and adaptive immune populations in vivo.

**CHIME does not alter immune reconstitution or responses**. To determine if the presence of Cas9 protein, the lentiviral sgRNA vector, or the process of transducing hematopoietic stem cells affected the reconstitution of immune cells, we generated chimeric mice using either mock-transduced WT LSK cells or Cas9-expressing LSKs that were transduced with a lentiviral sgRNA vector containing a non-targeting sgRNA. The stem cells were transduced at an MOI such that approximately half of the immune cells expressed the fluorescent reporter Vex, indicating the presence of the sgRNA vector (Supplementary Fig. 1e). We analyzed chimeras after immune reconstitution and found that the percentages of B cells, CD4[+] and CD8[+] T cells, CD11b[+] myeloid cells, and dendritic cells in the spleen were similar in WT and Cas9 + non-targeting sgRNA chimeras (Fig. 1d). To determine if the chimeric system altered the response of the immune system to a pathogen, we challenged chimeric mice (WT and Cas9 + non-targeting sgRNA as above) with LCMV Clone 13 virus and examined immune responses. WT and Cas9 + sgRNA chimeric mice had similar weight loss kinetics (Fig. 1e), indicating that the sgRNA delivery system did not alter the susceptibility of the mice to LCMV Clone 13. Serial viral titers in the blood and viral titers in the kidney at day 30 were comparable between WT and Cas9 + sgRNA chimeras, indicating a similar response to the viral infection (Fig. 1f and Supplementary Fig. 2a). These chimeras also had similar CD8[+] T cell responses as assessed by expression of T cell functional markers, coinhibitory receptors, and GP[33–41] antigen-specific responses (Supplementary Fig. 2b, 2c). These findings demonstrate that our Cas9-sgRNA delivery system does not affect immune reconstitution or immune responses to LCMV Clone 13 viral infection.

**Candidate genes are efficiently deleted in naive T cells**. To determine if CHIME could be used for studying genes in naive unperturbed T cells, we first examined whether this system altered T cell development. To examine this we analyzed thymic subsets in WT and Cas9 + non-targeting sgRNA chimeras and found no differences in the frequencies of double-negative (DN),

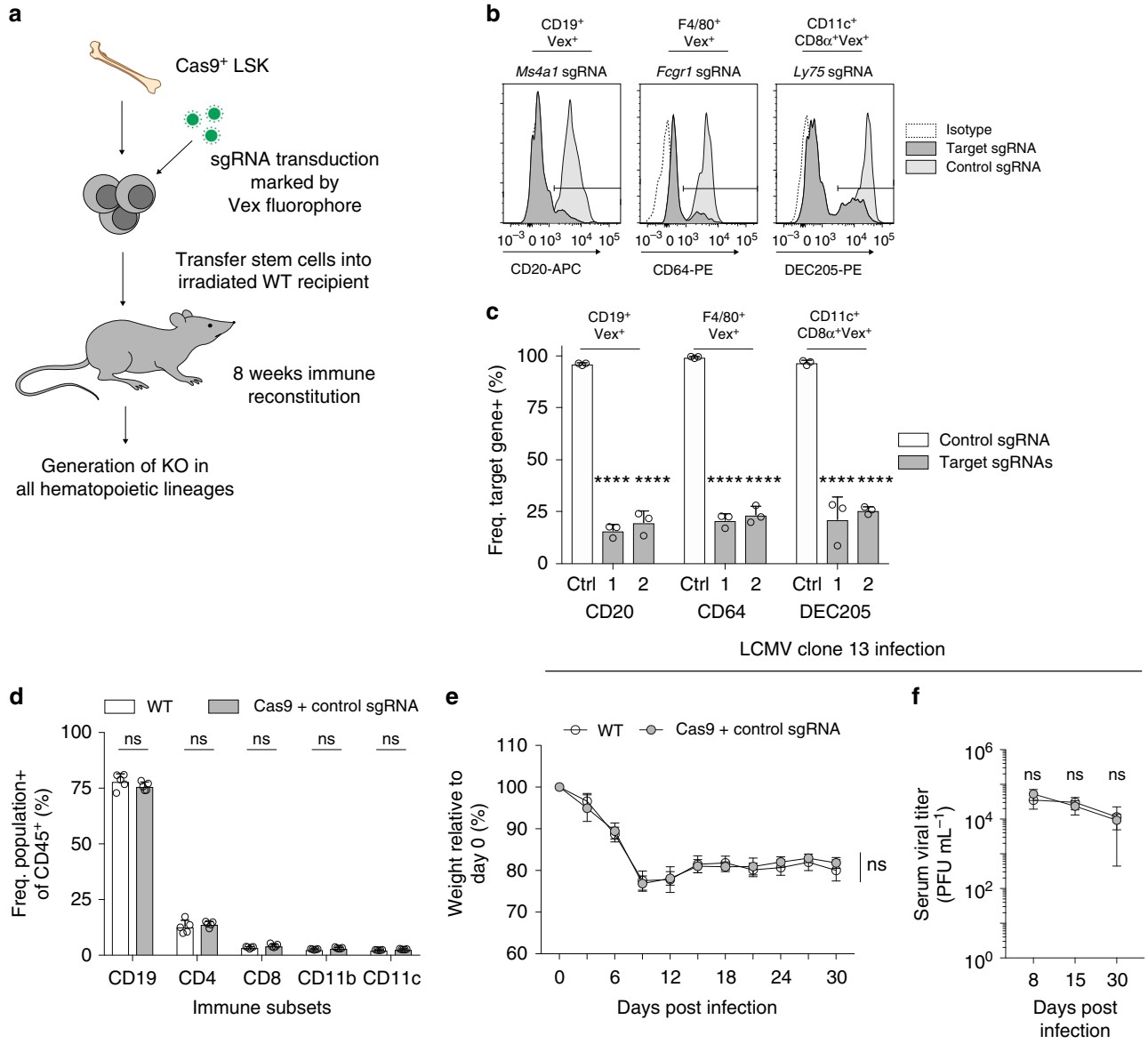

**Fig. 1** CHIME enables deletion of genes without impairing immune homeostasis. **a** Schematic of chimeric CRISPR-Cas9 system. **b** CD20 (left), CD64 (middle), or DEC205 (right) expression on B cells, macrophages, or dendritic cells, respectively from chimeric animals following transduction with a non-targeting control sgRNA or targeting sgRNAs to *Ms4a1*, *Fcgr1*, or *Ly75*. **c** Quantification of CD20, CD64, and DEC205 expression on relevant lineages from (**b**). **d** Comparison of frequencies of major immune lineages (of CD45+) in chimera mice (WT: WT stem cells mock transduced, Cas9 + sgRNA: Cas9 stem cells transduced with the Vex sgRNA expression vector) at homeostasis. **e**, **f** Chimeric mice were infected with $4 \times 10^6$ plaque-forming units (PFU) LCMV Clone 13 and their weight loss (e) and serum viral titer (f) were determined. All experiments had at least three biological replicate animals per group and are representative of two independent experiments. Bar graphs represent mean and error bars represent standard deviation. Statistical significance was assessed among the replicate bone marrow chimeras by one-way ANOVA (**c**, **d**), or two-way ANOVA (**e**, **f**) (*$p < .05$, **$p < .01$, ***$p < .001$, ****$p < .0001$). See also Supplementary Fig. 1 and 2. Source data are provided as a source data file

double-positive (DP), or CD4/CD8 single-positive (SP) populations in WT and Cas9 + non-targeting sgRNA chimeras (Fig. 2a). We also examined the naive status of CD8+ T cells from these chimeric mice and found no differences in CD44, CD62L, and CD69 percentages (Fig. 2b–d). To determine if CHIME enabled efficient deletion of genes in naive CD4+ and CD8+ T cells, we created chimeras carrying two non-targeting control sgRNAs or three *Pdcd1* targeting sgRNAs. We stimulated T cells from these chimeric mice with αCD3/CD28 to induce PD-1 expression and found a significant reduction of PD-1 expression in the presence of targeting sgRNAs but not control sgRNAs. On average 80% deletion was achieved in both CD4+ and CD8+ T cells (Fig. 2e, f). Analyses of naive CD4+ and CD8+ T cells from these mice prior

to stimulation using the TIDE assay[25] confirmed that these T cells had ~80% aberrant sequences, indicating efficient CRISPR-mediated indel formation (Fig. 2g and Supplementary Fig. 2d). We next analyzed off-target effects in this system by performing the TIDE assay on the top three predicted off-target sites for each of the three *Pdcd1* sgRNAs, and found minimal off-target editing above background in both CD4+ (Fig. 2h) and CD8+ T cells (Fig. 2i). Thus, this system can be used to knockout genes in naive T cells without altering their development or state.

**T cell-intrinsic functions can be examined in disease models.** To assess whether this Cas9-sgRNA delivery system can be used to

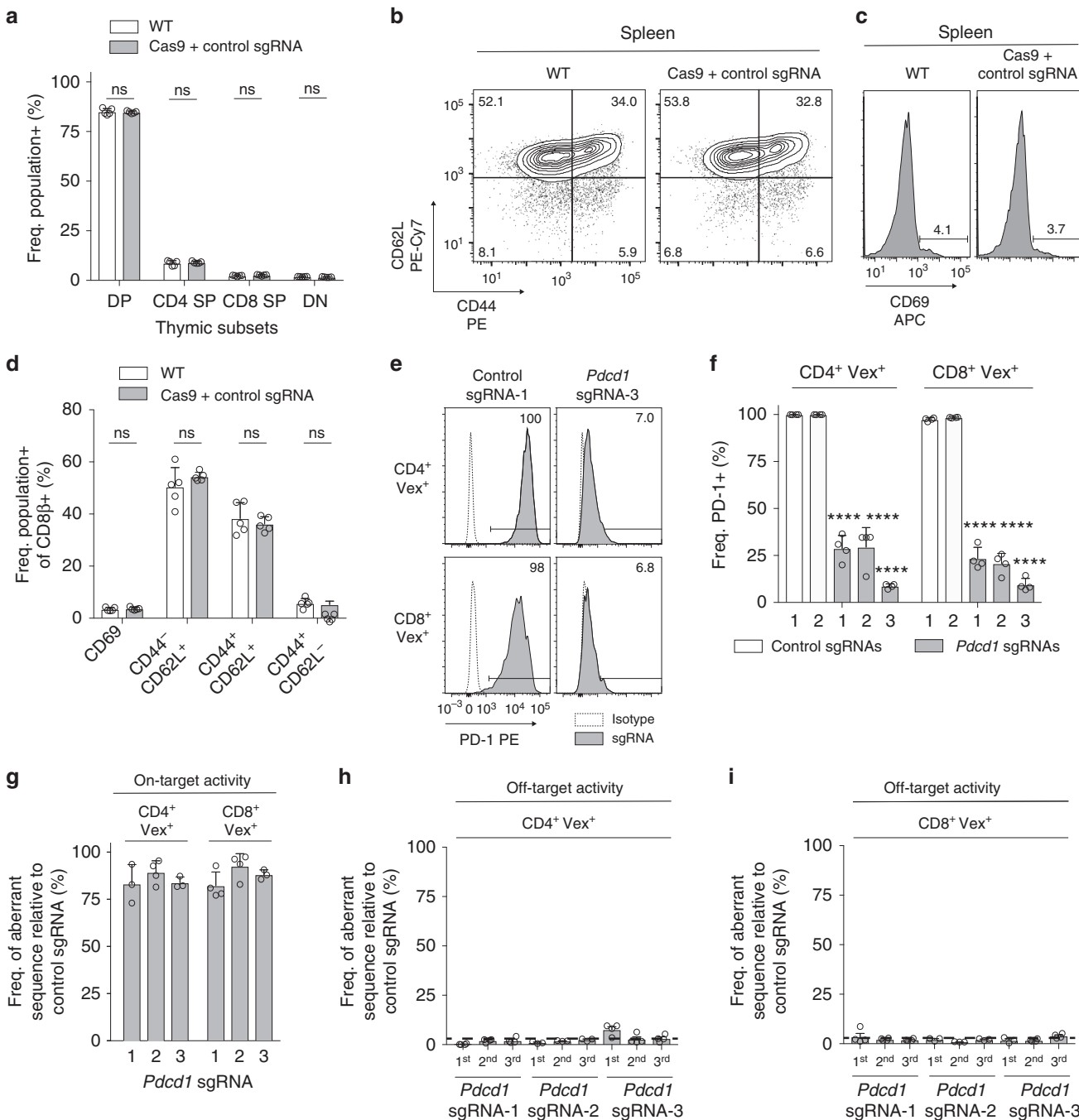

**Fig. 2** Deletion of genes in naive T cells with minimal off-target editing using CHIME. **a** Quantification of thymic subsets (CD4⁻ CD8⁻, CD4⁻ CD8⁺, CD4⁺ CD8⁻, CD4⁺ CD8⁺) from WT or Cas9 + control sgRNA chimeric mice. **b** Representative flow cytometry plots of CD44, CD62L, and **c** CD69 from splenic CD8⁺ T cells. **d** Quantification of naïve status of CD8⁺ T cells in (**b**, **c**). **e** Flow cytometry plots of PD-1 expression in CD4⁺ T cells (top panel) and CD8⁺ T cells (bottom panel) from representative non-targeting control sgRNA or *Pdcd1* sgRNA chimeras following αCD3/CD28 stimulation. **f** Quantification of PD-1 expression for two control and three *Pdcd1* sgRNAs from (**e**). **g** TIDE assay on naïve CD4⁺ and CD8⁺ T cells for three *Pdcd1* targeting sgRNAs. **h**, **i** TIDE assay on naïve (**h**) CD4⁺ and (**i**) CD8⁺ T cells designed to detect the top three predicted off-target sites (1st, 2nd, 3rd) for three *Pdcd1* targeting sgRNAs. Dashed line represents the aberrant sequence (%) when comparing two non-targeting control sgRNAs (background aberrant sequence). All experiments had at least three biological replicate animals per group and are representative of two independent experiments. Bar graphs represent mean and error bars represent standard deviation. Statistical significance was assessed among the replicate bone marrow chimeras by one-way ANOVA (**a**, **d**, **f**) (*$p < .05$, **$p < .01$, ***$p < .001$, ****$p < .0001$). See also Supplementary Fig. 2. Source data are provided as a source data file

identify intrinsic regulators of T cell function in CD8⁺ T cells, we used two models: LCMV Clone 13 viral infection as a model of T cell exhaustion, and MC38-OVA tumors as a model of tumor immunity. T cells responding to LCMV Clone 13 viral infection encounter multiple inhibitory mechanisms[26,27], many of which are conserved in the tumor microenvironment (TME)[17,28]. The MC38-OVA tumor model was used to directly assess antigen-specific T cell suppressive mechanisms in the TME. To analyze antigen-specific T cells in vivo, we used Cas9-expressing donor mice with the transgenic T cell receptors P14 (specific to the LCMV CD8

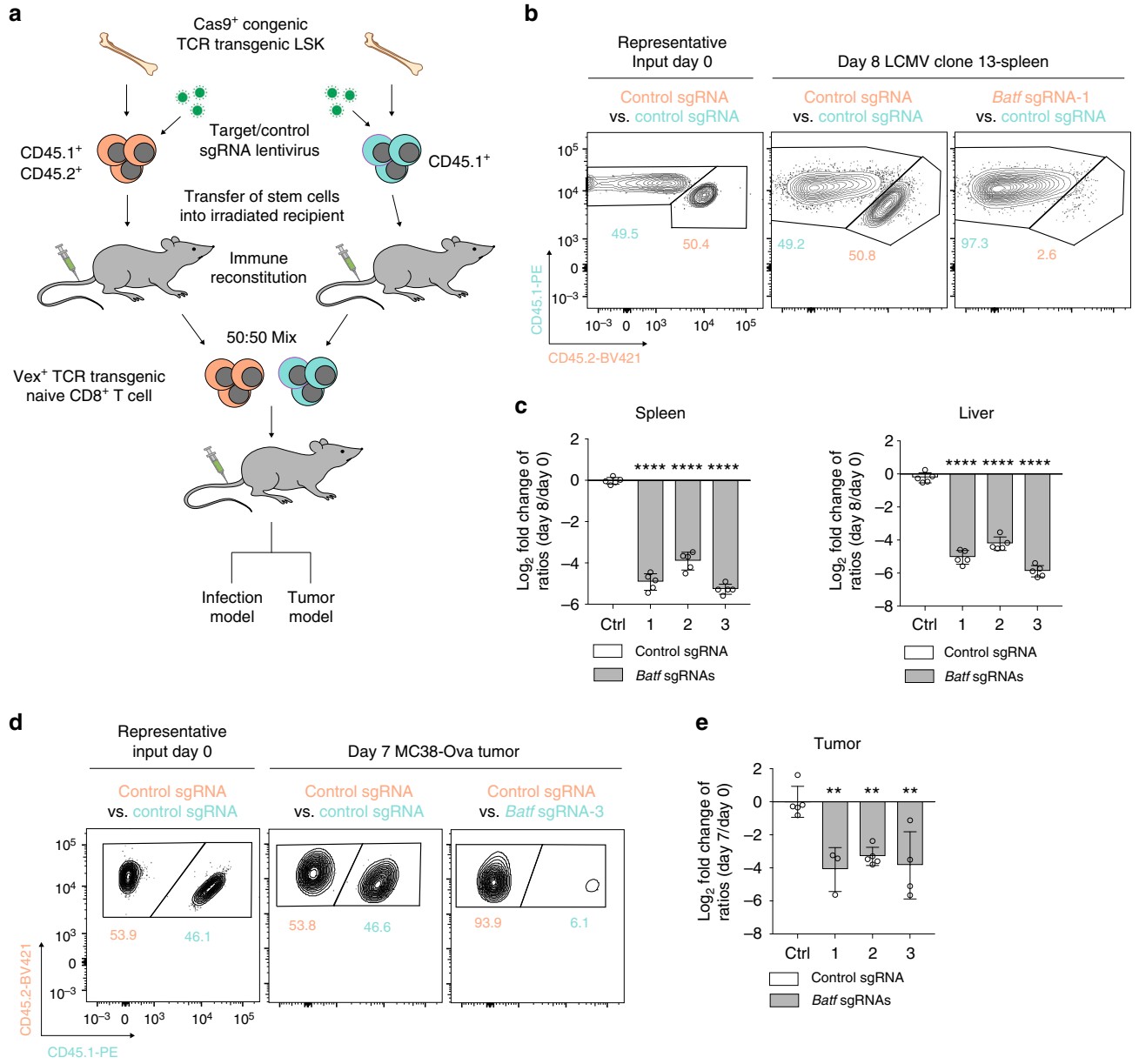

**Fig. 3** CHIME recovers known positive regulators of effector CD8+ T cell responses. **a** Schematic of competitive assays. **b** Representative input and output flow cytometry plots of P14 T cells containing control sgRNA vs. control or *Batf* sgRNAs in the spleen 8 days post LCMV Clone 13 viral infection. **c** Quantification of P14 T cells containing control or *Batf* sgRNAs from the spleen (left) and liver (right) in (**b**), normalizing the output flow cytometry plots to the input ratios at day 0 and log$_2$ transforming the data. **d** Representative output flow cytometry plots of OT-1 T cells containing control sgRNA vs. control or *Batf* sgRNAs in MC38-OVA tumors on day 7 post injection. **e** Quantification of OT-1 T cells containing control or *Batf* sgRNAs in (**d**) with normalization of the output flow cytometry plots to the input ratios at day 0 and log$_2$ transformation of the data. All experiments had three biological replicate animals per group and are representative of two independent experiments. Bar graphs represent mean and error bars represent standard deviation. Statistical significance was assessed among the replicate recipients of transferred T cells by one-way ANOVA (**c, e**) (*$p < .05$, **$p < .01$, ***$p < .001$, ****$p < .0001$). See also Supplementary Fig. 2. Source data are provided as a source data file

epitope GP$_{33-41}$) or OT-1 (specific to the ovalbumin CD8 epitope OVA$_{257-264}$). We transferred equal numbers (1:1 ratio) of congenically-marked antigen-specific gene-deleted naive CD8+ T cells and control cells to an unmanipulated host that was subsequently challenged with viral infection or tumor (Fig. 3a and Supplementary Fig. 2d) to compare the phenotype and function of the gene-deleted and control T cells in the same microenvironment.

We first evaluated the effect of deleting *Batf*, an essential transcription factor for effector T cell differentiation[19,29] during LCMV Clone 13 viral infection. Analysis of the *Batf* sgRNA-containing cells pre-transfer (input) by the TIDE assay indicated

that the indel percentage was on average 90% (Supplementary Fig. 2e). We infected recipient mice with LCMV Clone 13 and analyzed the ratio of control sgRNA-containing P14 T cells to control sgRNA or to *Batf* sgRNA-containing P14 T cells (Fig. 3b, c and Supplementary Fig. 2f) in the spleen on day 8 post infection. The ratios of P14 T cells with control sgRNA vs. control sgRNA remained unchanged compared with the input ratios (Fig. 3b). In contrast, P14 TCR transgenic T cells containing the *Batf* sgRNA were significantly depleted for three different *Batf* sgRNAs, which recapitulates both germline knockout and shRNA knockdown phenotypes of BATF in CD8+ T cells during LCMV

infection[19,29]. *Batf* sgRNA-containing T cells were similarly depleted in the liver of recipient mice, indicating a robust effect in multiple organs (Fig. 3c). Thus, the chimeric system can be used to evaluate the cell-intrinsic role of genes that regulate T cell differentiation and function in the LCMV Clone 13 infection model.

To determine if we could also examine the cell-intrinsic function of genetically perturbed CD8+ T cells in the TME, we transferred naive OT-1 transgenic CD8+ T cells into recipient mice and implanted MC38-OVA tumor cells. We again used sgRNAs targeting *Batf*, as an example of a gene deletion that reduced T cell expansion. The ratios of control sgRNA to control sgRNA-containing OT-1 T cells in the tumor did not change significantly at day 7 post-tumor implantation, compared with input (Fig. 3d, e and Supplementary Fig. 2g). In contrast, OT-1 T cells with the three *Batf* sgRNAs had a significantly reduced expansion compared with control cells (Fig. 3d, e). These results indicate that the chimera system can be used to perturb genes and evaluate cell-intrinsic effects in multiple disease models.

**Performing an in vivo CRISPR screen in T cells with CHIME.** We sought to determine if we could evaluate the function of multiple genes in a pooled fashion[30]. In order to determine the number of genetic perturbations that could feasibly be evaluated in a single animal, we performed bandwidth experiments to determine the number of integrated sgRNAs (barcodes) that could be successfully delivered to WT LSK cells and recovered from major immune populations (Supplementary Fig. 1b and Supplementary Fig. 3a–b). We transduced LSK cells with a pool of vectors expressing 20,000 distinct barcodes and determined the number of recovered barcodes in bone marrow, thymic, and splenic immune populations by sequencing (Supplementary Fig. 3c–d). The number of recovered barcodes varied considerably between cell types (least in thymic DP subset, and greatest in B cells), which is likely a consequence of the different differentiation paths from LSK progenitors (Supplementary Fig. 3d). In the spleen we recovered approximately 500 sgRNAs per mouse in all of the lineages analyzed (Supplementary Fig. 3d). Thus the number of recovered barcodes from many of the evaluated cell populations was sufficiently high to enable pooled in vivo screens using this system. Assuming recovery of 500 sgRNAs per mouse, we performed a Monte Carlo simulation to determine the number of pooled chimeras necessary to recover a 5000 sgRNA library. We found that theoretically pooling 30 chimeras enabled recovery of greater than 96% of a 5000 sgRNA library (Supplementary Fig. 3e).

We next designed a proof-of-concept pooled screen in effector CD8+ T cells to identify genes that regulate CD8+ T cell responses to LCMV Clone 13 infection. We generated 10 bone marrow chimeras from P14-transgenic Cas9-expressing CD45.1+ LSK cells transduced with a library of 110 sgRNAs targeting 21 genes relevant to T cell biology and 50 non-targeting control sgRNAs (Fig. 4a). The 21 genes were chosen to target three broad classes: TCR/cytokine signaling, coinhibitory/costimulatory receptors, and metabolism-associated. These classes were of particular interest given the important established roles of these pathways in T cell activation and immunosuppression during T cell responses. We confirmed that the recipient mice had a normal response to LCMV Clone 13 viral infection by assessing weight loss during the infection as well as kidney viral titer day 8 post infection (Supplementary Fig. 3f and 3g). We next analyzed CD44 expression, a T cell activation marker, on the CD45.1+ transferred population and confirmed that all transferred T cells containing the sgRNA library were CD44hi (Supplementary Fig. 3h, 3i), indicating all transferred cells were responding to LCMV Clone 13.

We recovered the majority of the 110 sgRNA pool from each LCMV-infected mouse (Supplementary Fig. 3j). Comparison of the 30 replicate mice revealed that sgRNA abundance (fold change relative to input) in both the spleen and lung were highly correlated (Pearson correlation coefficient 0.96), indicating high replicate animal reproducibility of library representation (Fig. 4b and Supplementary Fig. 3k). Further, we identified both enriched and depleted sgRNAs with approximately a normal distribution (Fig. 4c). The enrichment and depletion of sgRNAs was similar in spleen and lung, suggesting that the effect on CD8+ T cell differentiation was not tissue-dependent (Fig. 4d). We visualized the distribution of FDR values and identified marked depletion of sgRNAs targeting *Zap70*, *Cd28*, and *Dlat* (Fig. 4e, f, and Supplementary Table 1). We found that *Zap70* was the most significantly depleted gene in the screen, consistent with its essential role in TCR signaling. *Cd28* is a positive regulator of CD8+ T cell responses, and primary CD8+ T cell responses to LCMV are attenuated in CD28 deficient mice[31], supporting our screen results.

sgRNAs targeting *Kdr, Pdcd1, Adora2a,* and *Ptpn2* were markedly enriched in CD8+ T cells (Fig. 4g, h, and Supplementary Table 1). Enrichment of *Pdcd1* is consistent with its negative regulatory role in CD8+ T cell responses to LCMV Clone 13[32]. *Adora2a* and *Kdr* have both been implicated as negative regulators of CD8+ T cells in the TME[33,34], but have not been studied in the context of LCMV viral infection. The phosphatase *Ptpn2* was also an enriched hit, consistent with its established negative regulatory role in CD8+ T cells in autoimmunity and tolerance[35,36].

***Pdcd1* and *Ptpn2* negatively regulate anti-viral CD8+ T cells.** We focused on *Pdcd1* given its established negative regulatory role in the LCMV model[32] and on *Ptpn2* for validation studies. The role of *Ptpn2* in regulating CD8+ T cell responses to chronic viral infection has not been examined, although it is known to attenuate T cell responses to maintain tolerance and prevent autoimmunity[35,36].

To evaluate the role of *Pdcd1* in regulating anti-viral CD8+ T cell responses, we performed a 1:1 competitive assay with control sgRNA or *Pdcd1* sgRNA-containing P14 CD8+ T cells during LCMV Clone 13 viral infection. Using three sgRNAs, we found that *Pdcd1* sgRNA-containing P14 CD8+ T cells significantly outcompeted control sgRNA-containing cells in the spleen of infected recipient animals (Fig. 5a, b). Analysis of *Pdcd1* sgRNA-containing P14 CD8+ T cells following LCMV Clone 13 infection revealed significantly reduced PD-1 expression levels, as expected (Fig. 5c). To evaluate *Ptpn2*, we first confirmed efficient deletion (~80%) of *Ptpn2* using the TIDE assay (Fig. 5d) and by western blot (Fig. 5e). Using two sgRNAs, we found that *Ptpn2* sgRNA-containing P14 CD8+ T cells significantly outcompeted control sgRNA-containing cells in the spleen of infected recipient animals (Fig. 5f, g). These data indicate that both *Pdcd1* and *Ptpn2* have cell-intrinsic negative regulatory roles on CD8+ T cells in the LCMV Clone 13 infection model. In addition, this validates our screening approach for recovering (*Pdcd1*) and discovering (*Ptpn2*) regulators of T cell function in vivo.

**Discussion**
The discovery of new regulators of immune cell function using functional genomics has been limited by the difficulty of genetically perturbing immune cells without extensive ex vivo manipulation[19]. Here we describe CHIME, a chimera-based delivery system that enables rapid deletion of candidate genes in both innate (macrophages and dendritic cells) and adaptive (B cells, CD4+ or CD8+ T cells) immune populations without perturbing

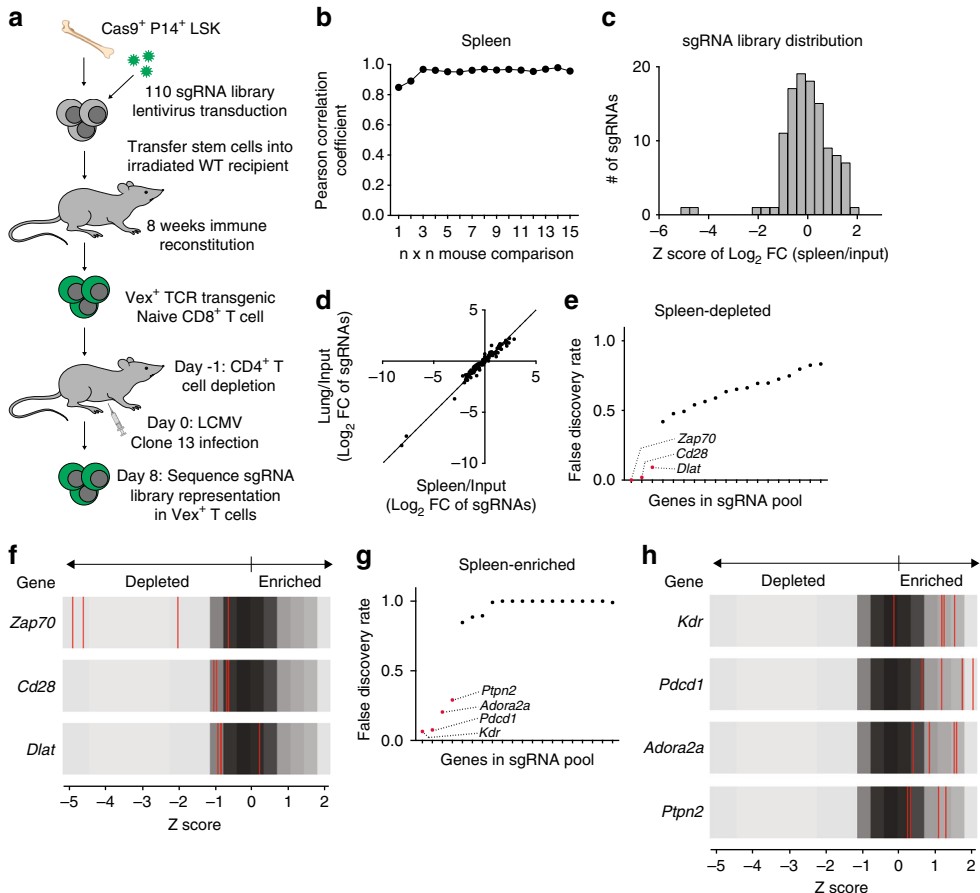

**Fig. 4** In vivo screening identifies regulators of CD8+ T cell responses to LCMV. **a** Schematic for in vivo screening. **b** Plot of Pearson R correlation vs. the number of replicate mice analyzed in the spleen from mice used in experiment in the screen. **c** Histogram depicting the distribution of the 110 sgRNAs relative to their log$_2$ fold change (spleen/input). **d** Correlation of lung/input log$_2$ fold change vs. spleen/input log$_2$ fold change. Dotted line depicts a perfect correlation ($y = x$). **e** Plot of the STARS calculated FDR for depleted in the spleen at day 8 post LCMV infection compared with input ($y$-axis) vs. the genes in the screening library ($x$-axis). Labeled genes are genes called as hits based on the FDR distribution. **f** Z-score plots of genes identified as depleted hits. Grey represents the background distribution of the library and red lines represent the four individual sgRNAs for the given genes. **g** Plot of the STARS calculated FDR for enriched in the spleen at day 8 post LCMV infection compared with input ($y$-axis) vs. the genes in the screening library ($x$-axis). Labeled genes are genes called as hits based on the FDR distribution. **h** Z-score plots of genes identified as enriched hits. Grey represents the background distribution of the library and red lines represent the four individual sgRNAs for the given genes. The in vivo screen had thirty biological replicate animals per group and is representative of one experiment. False discovery rate was assessed by the STARS software. See also Supplementary Fig. 3. Source data are provided as a source data file

cell states. In vivo analysis of gene function in immune populations through ES cell-targeted generation of knockout mice[37,38] is a lengthy process, while activation or cytokine stimulation of T cells to enable transduction[15] results in altered effector T cell differentiation[19]. Our system improves on available approaches by allowing deletion of genes in immune cell lineages in 8 weeks while maintaining normal immune development and function. Thus, we can rapidly analyze genes in immune cells in both physiological and disease contexts for the discovery of therapeutic targets. This system also enables high throughput screening using pooled sgRNA libraries[15,18,39] for the rapid identification and prioritization of new therapeutic targets. Our proof-of-concept screen with a 110 sgRNA library in conjunction with our bandwidth experiments suggest that this system enables screening of thousands of genes. In addition, our system expands the classes of immune cell lineages that can be screened, allowing efficient editing of macrophages and dendritic cells in vivo - two cell populations that are difficult to study in vivo without the use of knockout mice. Macrophages in particular have garnered attention as a source of cancer immunotherapy targets[40], and our system enables discovery of targets that alter macrophage

function. Moreover, our system expands the scope of phenotypes that can be evaluated by maintaining normal differentiation and homeostasis of the immune system. For example, our system makes it possible to target genes important for the earliest stages of T cell activation and differentiation, which has implications for developing new vaccine strategies. Our results suggest that CHIME is a versatile genetic platform can enable rapid discovery of therapeutic targets in immune cells in a spectrum of disease models.

## Methods

**Mouse breeding and production.** Seven- to ten-week-old female or male mice were used for all experiments and 7- to 14-week-old female or male mice were used as donors for bone marrow chimera experiments. WT C57BL/6 mice were purchased from The Jackson Laboratory. LoxP-STOP-LoxP Cas9 mice (B6J.129(B6N)-Gt(ROSA) 26Sortm1(CAG-cas9*,-EGFP)Fezh/J)[24] were a generous gift from Dr. Feng Zhang, Massachusetts Institute of Technology. These mice were bred to Zp3-Cre mice (C57BL/6-Tg(Zp3-cre)1Gwh/J) to delete the loxP-STOP-loxP in the female germline. The resulting Cas9-expressing strain was then bred to OT-I (C57BL/6-Tg(TcraTcrb)1100Mjb/J) or P14 (Taconic B6.Cg-Tcratm1Mom Tg (TcrLCMV)327Sdz backcrossed 10 generations to Jackson C57BL/6 J) TCR transgenic mice on the CD45.1 (B6.SJL-Ptprca Pepcb/BoyJ) congenic background. All strains used were backcrossed at least 10 generations to Jackson C57BL/6J.

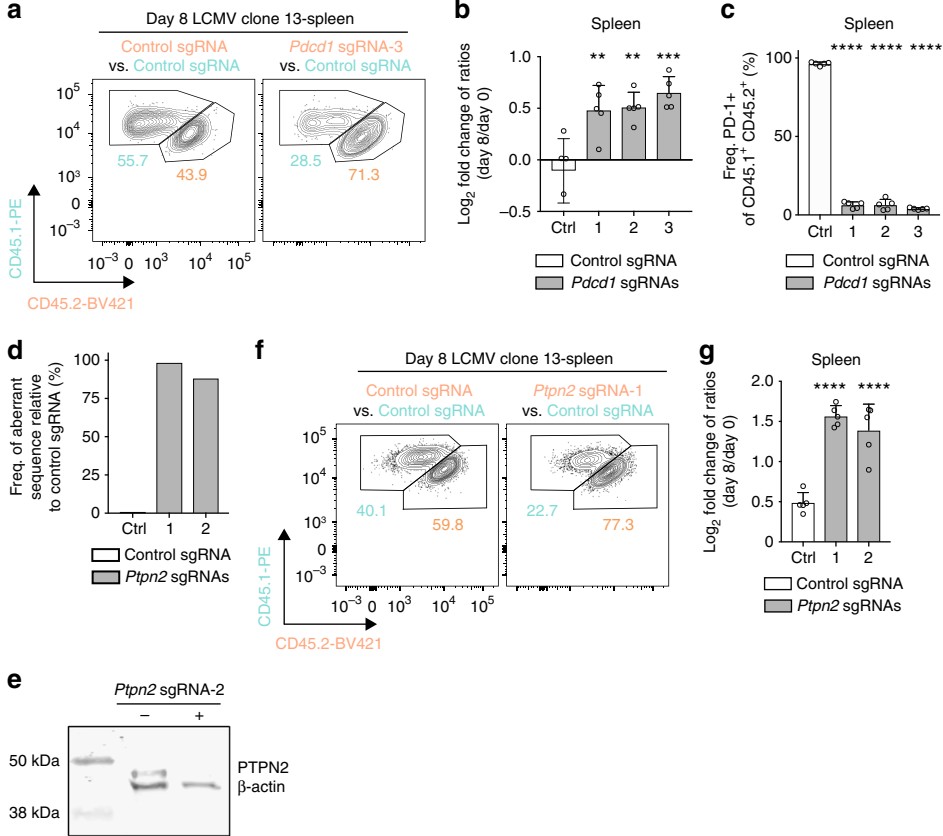

**Fig. 5** *Pdcd1* and *Ptpn2* are negative regulators of CD8[+] T cell responses to LCMV. **a** Representative output flow cytometry plots of P14 T cells containing control sgRNA vs. control or *Pdcd1* sgRNAs in the spleen day 8 post LCMV Clone 13 viral infection. **b** Quantification of **(a)** with normalization of the output flow cytometry plots to the input ratios at day 0 and $\log_2$ transformation of the data. **c** Quantification of PD-1 expression on P14 T cells containing control sgRNA or *Pdcd1* sgRNAs in the spleen day 8 post LCMV Clone 13 viral infection. **d** TIDE assay on naive CD8[+] T cells containing control sgRNA or *Ptpn2* sgRNAs. **e** Cropped western blot of splenic CD8[+] T cells from control sgRNA or *Ptpn2* sgRNA-containing chimeras. **f** Representative output flow cytometry plots of P14 T cells containing control sgRNA vs. control or *Ptpn2* sgRNAs in the spleen day 8 post LCMV Clone 13 viral infection. **g** Quantification of **(f)** with normalization of the output flow cytometry plots to the input ratios at day 0 and $\log_2$ transformation of the data. All experiments (except western blot) had five biological replicate animals per group and are representative of two independent experiments. Western blot had two pooled mice per group and is representative of three independent experiments. Bar graphs represent mean and error bars represent standard deviation. Statistical significance was assessed among the replicate recipients of transferred T cells by one-way ANOVA (**b**, **c**, **g**) (*$p < .05$, **$p < .01$, ***$p < .001$, ****$p < .0001$). Source data are provided as a source data file

Sample size was chosen to ensure the possibility of statistical analysis and to also minimize the use of animals. Data exclusion was not used. Age and sex-matched animals were used for each experiment. Animals were also co-housed when possible. All attempts to reproduce our findings were successful. The LCMV Clone 13 infection experiment (Fig. 1e, f) was blinded during data collection. All experimental mice were housed in specific pathogen-free conditions and used in accordance with the ethical regulations and animal care guidelines pre-approved by the Harvard Medical School Standing Committee on Animals and the National Institutes of Health.

**Guide RNA design and cloning**. The sgRNA oligonucleotides (sequences in Supplementary Table 2), were designed using the Broad CRISPR algorithm[41]. Off-target sites were identified using the Benchling CRISPR design tool which incorporates off-target rules from the MIT CRISPR algorithm[42]. sgRNAs were cloned into our sgRNA vector using a BsmBI restriction digest. This sgRNA vector was created by modifying an existing lentiviral shRNA vector[19]. Briefly, the modified vector contains the human U6 promoter (with Lac operator site) for expression of a sgRNA as well as the human PGK promoter for expression of the fluorophore Vex. The plasmid and full sequence have been deposited on Addgene (Name: pXPR_053, Addgene ID: 113591).

**Bone marrow isolation and chimera setup**. Femurs, tibias, hips, and spines were isolated from CD45.1[+] donor mice, crushed, and ACK-lysed. LSK (lineage[−] Sca-1[+] Kit[+]) were enriched with a CD117 MACS isolation kit and then sorted to purity. The LSK were spin transduced with lentiviral constructs on a Retronectin-coated plate. LSK cells were then transferred intravenously into irradiated CD45.2[+] recipients.

**Cell lines**. MC38-OVA cells were cultured in DMEM supplemented with 10% FBS, 1% penicillin/streptomycin, and 20 µg/ml gentamicin. MC38-OVA cells (gift from Natalie Collins, Dana Farber Cancer Institute) were produced by transduction of parental MC38 cells (gift from Dario Vignali, University of Pittsburgh) with the lentiviral vector TRC-pLX305 (Broad Institute) containing ovalbumin (OVA) protein. MC38-OVA cells were selected for 2 days with 2 µg/mL puromycin prior to use. MC38-OVA cells were validated based on expression of a selectable marker and exome sequencing. BHK-21 cells (gift from E. John Wherry, University of Pennsylvania) were cultured in DMEM supplemented with 10% FBS, 1% penicillin/streptomycin, and 5% tryptose phosphate broth. Vero cells (gift from E. John Wherry, University of Pennsylvania) were cultured in EMEM supplemented with 10% FBS, and 1% penicillin/streptomycin. 293x cells (gift from Cigall Kadoch, Dana Farber Cancer Institute) were cultured in DMEM supplemented with 10% FBS, 1% penicillin/streptomycin, and 20 µg/ml gentamicin. 293x cells (HEK variant) were used for making lentivirus, which was validated by titering. All cell lines were confirmed to be mycoplasma negative.

**In vitro stimulation**. To analyze PD-1 expression by flow cytometry, sorted naïve CD4[+] or CD8[+] T cells were stimulated with 4 µg/mL αCD3/CD28 (BioXCell Cat# BE0001, BE0015) for 72 h. Cells were then stained and analyzed by flow cytometry.

**TIDE assay**. The TIDE (Tracking of Indels by DEcomposition) assay[25] was performed on isolated naive T cells. To do this, DNA was extracted from cells (Qiagen DNeasy kit) and a 500 base pair product spanning the sgRNA cut site amplified by PCR. This PCR product was purified (Qiagen PCR Purification kit) and analyzed by Sanger sequencing. All TIDE primers are listed in Supplementary Table 2.

**Adoptive T cell transfer**. Spleens were isolated from chimeric mice (>8 week post reconstitution) and naïve CD8$^+$ T cells were purified using a naïve CD8$^+$ MACS kit (>95% purity). Cells were stained with lineage-specific antibodies (TER-119, B220, and Gr-1) and 7-Aminoactinomycin D (7-AAD) and then sorted (Lineage$^-$, 7-AAD$^-$, Vex$^+$ cells). For LCMV studies, cells were transferred (500:500 mix) to recipient mice on day -1, and mice were infected with LCMV Clone 13 (as below) on day 0. For tumor studies, cells were transferred (1000:1000 mix) to recipient mice on day -1, and mice were injected with MC38-OVA (as below) on day 0.

**LCMV production and plaque assay**. LCMV Clone 13 virus was produced by infecting BHK-21 cells with an LCMV Clone 13 virus stock at an MOI of 0.03 and harvesting viral supernatants 48 h later. Viral titers were determined by plating diluted viral stocks or serum/tissue samples on Vero cells with an agarose overlay. Four days later the Vero cells were stained with neutral red dye, and 14 h later plaques quantified.

**LCMV infection and analysis**. Mice were infected with $4 \times 10^6$ PFU LCMV Clone 13 i.v., monitored for weight loss, and bled or sacrificed at day 8, 15, or 30 post infection for flow cytometry analyses. For viral titer studies, mice were bled at days 8, 15, and 30 post infection. Liver lymphocytes were isolated by dissociation of the liver followed by a 40%/60% Percoll gradient. Lung lymphocytes were isolated by dissociation of the lung on a gentleMACS Dissociator followed by a 37 °C incubation in collagenase for 30 min. Lymphocytes were enriched on a 40%/60% Percoll gradient. To deplete CD4$^+$ T cells, mice were injected i.p. with 200 μg αCD4 (BioXCell Cat# BE0003-1) on days -1 and 1 (relative to LCMV Clone 13 injection on day 0).

**Tumor injection**. Mice were anesthetized with 2.5% 2,2,2-Tribromoethanol (Avertin) and injected in the flank subcutaneously with $2 \times 10^6$ MC38-OVA tumor cells. Tumors were measured every 2–3 days once palpable using a caliper. Tumor volume was determined by the volume formula for an ellipsoid: $1/2 \times D \times d^2$ where D is the longer diameter, and d is the shorter diameter. Mice were sacrificed when tumors reached 2 cm$^3$ or upon ulceration.

**Tumor infiltrating lymphocyte isolation**. Tumors were excised and dissociated using a gentleMACS Dissociator. Tumors were then incubated in collagenase for 20 min at 37 °C. Lymphocytes were enriched using a 40%/70% Percoll gradient.

**Flow cytometry and cell sorting**. Flow cytometry analyses were performed on a BD LSR II or BD FACSymphony A5 and cell sorting was performed on a BD Aria IIu. Antibodies and dyes used include: 7-AAD BD Biosciences Cat# 559925 (1:100 dilution), Ki67-PerCP-Cy5.5 BD Biosciences Cat# 561284 (1:100 dilution), B220 Biolegend Cat# 103208, 103326 (1:100 dilution), CD11b Biolegend Cat# 101208, 101216 (1:100 dilution), CD11c Biolegend Cat# 117307, 117328 (1:100 dilution), CD127 Biolegend Cat# 135014, 135024 (1:100 dilution), CD19 Biolegend Cat# 115533 (1:100 dilution), CD20 Biolegend Cat# 150412 (1:100 dilution), CD25 Biolegend Cat# 101904 (1:100 dilution), CD3ε Biolegend Cat# 100220, 100308, 100336 (1:100 dilution), CD4 Biolegend Cat# 100516, 100531, 100543 (1:100 dilution), CD44 Biolegend Cat# 103008, 103028, 103030 (1:100 dilution), CD45.1 Biolegend Cat# 110708, 110716 (1:100 dilution), CD45.2 Biolegend Cat# 109824, 109832 (1:100 dilution), CD49b Biolegend Cat# 108909 (1:100 dilution), CD5 Biolegend Cat# 100608 (1:100 dilution), CD62L Biolegend Cat# 104417 (1:100 dilution), CD64 Biolegend Cat# 139303 (1:100 dilution), CD69 Biolegend Cat# 104513 (1:100 dilution), CD8α Biolegend Cat# 100737 (1:100 dilution), CD8β Biolegend Cat# 126606, 126608, 126610, 126620 (1:100 dilution), c-Kit Biolegend Cat# 135108 (1:100 dilution), F4/80 Biolegend Cat# 123116 (1:100 dilution), Gr-1 Biolegend Cat# 108408 (1:100 dilution), Granzyme B Biolegend Cat# 515403 (1:100 dilution), NK1.1 Biolegend Cat# 108708, 108732 (1:100 dilution), PD-1 Biolegend Cat# 135206, 135209 (1:100 dilution), Sca-1 Biolegend Cat# 108108, 108128 (1:100 dilution), TCR Vα2 Biolegend Cat# 127814 (1:100 dilution), TCR Vβ5 BD Biosciences Cat# 562087 (1:100 dilution), Ter-119 Biolegend Cat# 116208 (1:100 dilution), Tim-3 Biolegend Cat# 119703, 119723 (1:100 dilution), TruStain fcX Biolegend Cat# 101320 (1:50 dilution), Rat IgG2a κ Isotype Biolegend Cat# 400508 (1:100 dilution), Rat IgG2b κ Isotype Biolegend Cat# 400612 (1:100 dilution), and Near-IR Fixable Live/Dead Thermo Fisher Scientific Cat# L34976 (1:500 dilution). GP$_{33–41}$ tetramer was obtained from the NIH Tetramer Core Facility and used at a 1:400 dilution. Cross-linking and depleting antibodies (CD3ε, CD28, CD4) were purchased from BioXCell.

**Western blotting**. Spleen and lymph nodes (cervical, axillary, inguinal) were isolated from control or Ptpn2-2 sgRNA-containing chimeric mice. Spleen and lymph node were pooled and CD8$^+$ T cells were enriched using CD8α microbeads. The spleen and lymph node samples were then sorted for CD8β$^+$ Vex$^+$. Whole cell lysates were generated using a mixture of Pierce RIPA buffer and protease/phosphatase inhibitor at a final concentration of 1 mg/mL. Protein concentration was measured with a BCA protein assay kit. Subsequently, 30 μg of protein was run on a NuPage 4–12% bis-tris protein gel and then transferred to a nitrocellulose

membrane. The membrane was incubated overnight in Odyssey blocking buffer followed by staining with anti-TC-PTP (C term) mouse monoclonal IgG antibody (Medimabs Cat# MM-0019)[43] and anti-β actin rabbit polyclonal IgG antibody (Abcam Cat# ab8227) at a 1:1000 dilution for 1 h at room temperature. The membrane was washed with TBS-T and then incubated with secondary antibodies IRDye 680RD Goat anti-Mouse IgG (LI-COR Biosciences Cat# 925-68070) and RDye 800CW Donkey anti-Rabbit IgG (H + L) (LI-COR Biosciences Cat# 925-32213), at a 1:10000 dilution for 1 h at room temperature. The membrane was washed and visualized using the Li-Cor Clx Imaging System (Li-Cor). Full uncropped blot is available in the source data file.

**Bandwidth and _in vivo_ screening experiments**. For bandwidth experiments a 20,000 sgRNA library (Brie library from the Genetic Perturbation Platform at the Broad Institute) was used to transduce WT OT1$^+$ LSK, and bone marrow chimeras were created as described above. Following immune reconstitution, immune subsets were sorted based on lineage markers and Vex. DNA was isolated using the Qiagen DNeasy kit, the sgRNA region PCR amplified, and sequenced on an Illumina HiSeq to determine sgRNA abundance. Recovery of sgRNAs was assessed using a 25 read cutoff. For calculations of the number of chimeras needed for recovery of a 5000 sgRNA library we performed a Monte Carlo simulation where we sampled 500 sgRNAs with replacement from a pool of 5000 sgRNAs. We determined the number of samplings of 500 sgRNAs it would take to recover >95% of the 5000 sgRNA library with a probability of >95%. This simulation was performed 10,000 times and indicated that a library of 5000 sgRNAs could theoretically be recovered by pooling 30 chimeras, assuming recovery of 500 sgRNAs per mouse. For screening experiments, a 110 sgRNA library was created consisting of 4 sgRNA/gene (for 21 experimental target genes), 3 sgRNA/gene (for 2 predicted irrelevant target genes), and 20 predicted non-targeting control sgRNA (no known target). Experimental genes were curated from the literature and consisted of 3 broad classes: TCR/cytokine signaling, coinhibitory/costimulatory receptors, and metabolism-associated. P14$^+$ Cas9$^+$ CD45.1$^+$ LSK were transduced with this library at an infection rate of 35% and were used to create bone marrow chimeras. An infection rate of 35% was chosen based on a Poisson distribution such that the majority of our transduced cells were single integrants. Following immune reconstitution, naive CD8$^+$ T cells were isolated using a Miltenyi naive CD8 negative selection kit and sorted for Vex$^+$. 5,000 Vex$^+$ naive CD8$^+$ T cells were transferred to 30 recipient animals (150,000 cells total which equates to >1000X coverage of the 110 sgRNA library) that were infected the next day with LCMV Clone 13 as described. 1.5 million (500,000 in technical triplicate) of the Vex$^+$ naive CD8$^+$ T cells were kept for input sequencing. At 8 days post LCMV infection, spleens were isolated from recipient animals and CD8$^+$ T cells enriched with a Miltenyi CD8 positive selection kit followed by sorting for CD8$^+$ CD45.1$^+$ CD44$^+$ cells. In addition, 8 days post LCMV infection, lungs were isolated from recipient animals and lymphocytes enriched as above followed by sorting for CD8$^+$ CD45.1$^+$ CD44$^+$ cells. DNA was isolated using the Qiagen DNeasy kit, the sgRNA region PCR amplified, and sequenced on an Illumina HiSeq to determine sgRNA abundance. Enriched or depleted genes were ranked using the STARS software[41] and associated FDR. Visualization of this allowed us to identify a clear gap in the FDR distribution between top depleted/enriched genes and the rest of the pooled genes, which we called as potential depleted/enriched hits. For visualization of sgRNAs for individual genes, log$_2$ fold changes (spleen/input) were z-scored in MATLAB and individual sgRNAs called out in red on top of the distribution of all other sgRNAs. To confirm normal LCMV Clone 13 infection dynamics weight loss was monitored, and kidneys were isolated day 8 post LCMV infection for quantification of viral titer as above.

**Statistical analysis**. Statistical analyses were performed using GraphPad Prism 7 software. The data were considered statistically significant with p values <0.05 by unpaired Student's t test for comparing two groups, one-way ANOVA for single comparisons with groups greater than two, two-way ANOVA for repeated measures comparisons or for multiple comparisons within groups, the Pearson R test for pooled screen quality control, and the STARS software for evaluating screen hits.

**Reporting summary**. Further information on experimental design is available in the Nature Research Reporting Summary linked to this article.

## Data availability
The data and materials that support the findings of this study are available from the corresponding author upon reasonable request. Source data underlying graphs in Figs. 1–5 and Supplementary Fig. 2 and 3 has been provided as a source data file. Plasmid and full sequence have been deposited on Addgene (Name: pXPR_053, Addgene ID: 113591).

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

## Acknowledgements
We thank Natalie Collins, Cigall Kadoch, and Dario Vignali for sharing cancer cell lines; Feng Zhang for sharing LoxP-STOP-LoxP-Cas9 mice; Kristen Pauken, Makoto Kurachi, and E. John Wherry for providing MSCV-Vex plasmid, LCMV Clone 13 viral stocks, BHK/Vero cell lines, and technical advice. We also thank the Dana-Farber/Harvard Cancer Center DNA Resource Core for Sanger sequencing. This work was supported by funding from a Broad SPARC grant, a Novartis STEM grant, T32CA207021 to M.W.L., T32GM007753 to S.A.W., and U19AI133524 to A.H.S and W.N.H.

## Author contributions
M.W.L., W.N.H. and A.H.S. conceived the project and wrote the manuscript with assistance from T.H.N., M.A.C., and J.E.G.; M.W.L., T.H.N., M.A.C., W.N.H. and A.H.S. designed experiments; M.W.L., T.H.N. and M.A.C. performed and analyzed all experiments with assistance from K.B.Y., J.D.T., F.D.B., and J.E.G. J.G.D. helped with bandwidth/screen library construction, experimental design, sequencing, and computational analysis of results. S.A.W. helped with analysis and visualization of screen data. D.J.C. helped with computational analyses of bandwidth experiments.
