## [Peer Review File · Nature Communications]

Reviewers' Comments:

Reviewer #1:

Remarks to the Author:

The manuscript submitted by LaFleur et al. describes in detail the means to use Cas9 and sgRNAs to effectively knock out genes in hematopoietic lineages including lymphocytes, macrophages, and dendritic cells. The experimental design and controls were appropriate for the studies, and the presentation makes a compelling case for using such a technique for the identification of genes involved in immune regulation. In reconstruction experiments the investigators showed that they could simultaneously screen 110 sgRNAs for depletion or enrichment. They describe being able to recover 500 sgRNAs per mouse from a 20,000 library, and then assert that they could thus screen 5000 sgRNAs by pooling 30 chimeras. I didn't quite understand that calculation, but I suppose they are factoring in some error.

Despite the elegance of the studies, and the fact that they actualized what everyone assumed to be possible, they didn't actually break any new ground in this study. The use of lentiviral transduction to express genes in bone marrow is well known. The use of sgRNAs to efficiently knock out genes in various mature cells including T cells is now established. And the use of screens for depletion or enrichment is established. Unfortunately, thus far, they have not discovered any genes previously not known to effect T cells responses.

Nonetheless, this is a nice demonstration of the power of CAS9 gene targeting and screening for genes that can be said to overcome the expense and time needed to perform forward genetics in mice. Although none of the components of the paper are novel, together they compelling in establishing a work flow for gene discovery. I can imagine that this work would provide confirmation that such studies are feasible and well worth pursuing.

Reviewer #2:

Remarks to the Author:

The article by LaFleur et al. demonstrates elegantly the usage of the CRISPR/Cas9 system for analysing/ understanding immune function in vivo. Therefore they are using LSK cells derived from a Cas9 tg mouse, which they transduce with sgRNAs in vitro to then transfer into lethally irradiated mice to make up their immune system from the manipulated cells. They nicely demonstrate the KO of different genes in different immune lineages before going on to show that the presence of a control sgRNA does not impair any lineage development. In the final experiments they use a targeted and "mini" screen approach to identify a novel regulator of CD8 T cell function. Overall this is a very good study, which demonstrates the usefulness of the CRISPR/ Cas9 system for investigating immune function. I have only a few points, which should be addressed before the manuscript can be accepted.

- In the abstract it is stated "without prior ex vivo manipulation". I am a bit confused by this, as the LSKs are manipulated by the sgRNA lentis.

- The authors should also mention in the Introduction a study by Chu VT, PNAS 2016 and Janic A, NATURE Med. 2018 in which the Cas9 tg mice were used to study immune gene function ex vivo and p53 biology in vivo, respectively.

- In Figure 4 a and b, the representation of sgRNAs/ barcodes is demonstrated by infecting the LSKs with 20000 sgRNAs. Why do you find less barcodes in the Stem cells than in the mature splenic cells? Shouldn't it be the other way around, if you consider the LSK population being the one being transduced with the 20000 sgRNAs?

- In the same experiment (Figure 4a,b) the conclusion in the text is that you need about 30

chimaeric mice to have a sgRNA representation of 5000. How did the authors come to this number?

- In order to have an exact number on how many guides are still detectable, i.e. the max number of guides per library which can be used, the authors could have done a dilution in number of sgRNAs transducing the LSKs, i.e. 20000, 10000, 5000, Since this is probably beyond the scope of this study I would recommend to add a few more detailed calculations based on the data they have. Additionally, I would recommend to put this data into the Supp Data, as it breaks the flow of the manuscript a bit.

- In the methods part it is mentioned that the library infection of LSKs was 35%. Why was that number chosen?

- The focused screen is very nice and identifies known and new CD8 T cell regulators. However, I would like to know a little bit more about why these 21 genes were chosen? There is one sentence, but this could be discussed a little bit more.

-Could the authors also state in more detail how the statistics were calculated for the animal experiments? Is it based on recipients or based on bonemarrow donors.

REVIEWERS' COMMENTS:

Reviewer #1 (Remarks to the Author):

The manuscript submitted by LaFleur et al. describes in detail the means to use Cas9 and sgRNAs to effectively knock out genes in hematopoietic lineages including lymphocytes, macrophages, and dendritic cells. The experimental design and controls were appropriate for the studies, and the presentation makes a compelling case for using such a technique for the identification of genes involved in immune regulation. In reconstruction experiments the investigators showed that they could simultaneously screen 110 sgRNAs for depletion or enrichment. They describe being able to recover 500 sgRNAs per mouse from a 20,000 library, and then assert that they could thus screen 5000 sgRNAs by pooling 30 chimeras. I didn't quite understand that calculation, but I suppose they are factoring in some error.

Despite the elegance of the studies, and the fact that they actualized what everyone assumed to be possible, they didn't actually break any new ground in this study. The use of lentiviral transduction to express genes in bone marrow is well known. The use of sgRNAs to efficiently knock out genes in various mature cells including T cells is now established. And the use of screens for depletion or enrichment is established. Unfortunately, thus far, they have not discovered any genes previously not known to effect T cells responses.

Nonetheless, this is a nice demonstration of the power of CAS9 gene targeting and screening for genes that can be said to overcome the expense and time needed to perform forward genetics in mice. Although none of the components of the paper are novel, together they compelling in establishing a work flow for gene discovery. I can imagine that this work would provide confirmation that such studies are feasible and well worth pursuing.

We thank reviewer 1 for these comments on our manuscript. We have provided additional information to clarify our calculations for screening 5000 sgRNAs by pooling 30 chimeras. We calculated this value using a Monte Carlo simulation and have clarified the text (methods section and Supplementary Figure 3) to explain this calculation.

Reviewer #2 (Remarks to the Author):

The article by LaFleur et al. demonstrates elegantly the usage of the CRISPR/Cas9 system for analysing/ understanding immune function in vivo. Therefore they are using LSK cells derived from a Cas9 tg mouse, which they transduce with sgRNAs in vitro to then transfer into lethally irradiated mice to make up their immune system from the manipulated cells. They nicely demonstrate the KO of different genes in different immune lineages before going on to show that the presence of a control sgRNA does not impair any lineage development. In the final experiments they use a targeted and "mini" screen approach to identify a novel regulator of CD8 T cell function. Overall this is a very good study, which demonstrates the usefulness of the CRISPR/ Cas9 system for investigating immune function. I have only a few points, which should be addressed before the manuscript can be accepted.

- In the abstract it is stated "without prior ex vivo manipulation". I am a bit confused by this, as the LSKs are manipulated by the sgRNA lentis.

We agree that this statement needs to be clarified as we are referring to ex vivo manipulation of mature differentiated immune lineages. We have edited the text to address this issue.

- The authors should also mention in the Introduction a study by Chu VT, PNAS 2016 and Janic A, NATURE

Med. 2018 in which the Cas9 tg mice were used to study immune gene function ex vivo and p53 biology in vivo, respectively.

We have added these references to the introduction.

- In Figure 4 a and b, the representation of sgRNAs/ barcodes is demonstrated by infecting the LSKs with 20000 sgRNAs. Why do you find less barcodes in the Stem cells than in the mature splenic cells? Shouldn't it be the other way around, if you consider the LSK population being the one being transduced with the 20000 sgRNAs?

We thank the reviewer for this question. We believe this is the result of differentiation of the LSK into progenitor cells and mature immune lineages to fill the hematopoietic niche following irradiation. This concept is supported by studies such as Grinenko et al. Nature Communications 2018 and Lu et al. Nature Biotechnology 2012. Thus, we believe that at one point every barcode was present in LSK cells but following differentiation of the LSK these barcodes are now present in the mature cells, but not LSK.

- In the same experiment (Figure 4a,b) the conclusion in the text is that you need about 30 chimaeric mice to have a sgRNA representation of 5000. How did the authors come to this number?

As explained in response to Reviewer 1, we calculated this value using a Monte Carlo simulation which we have included in the revised methods section and Supplementary Figure 3. We have clarified the text to explain this calculation.

- In order to have an exact number on how many guides are still detectable, i.e. the max number of guides per library which can be used, the authors could have done a dilution in number of sgRNAs transducing the LSKs, i.e. 20000, 10000, 5000, Since this is probably beyond the scope of this study I would recommend to add a few more detailed calculations based on the data they have. Additionally, I would recommend to put this data into the Supp Data, as it breaks the flow of the manuscript a bit.

We thank the reviewer for these thoughtful comments. Here we used a 20000 sgRNA library assuming the level of complexity was sufficiently high that we could count detectable sgRNAs following reconstitution. To improve the flow of the manuscript we have moved the bandwidth data to the supplement.

- In the methods part it is mentioned that the library infection of LSKs was 35%. Why was that number chosen?

The infection rate of 35% (based on a range of 30-40%) was chosen such that based on a Poisson distribution the majority of our cells were untransduced, and the majority of the remaining cells were single integrants. This approach allowed us to avoid multiple integrations while still infecting a substantial proportion of our LSK. We have expanded on this point in the methods section.

- The focused screen is very nice and identifies known and new CD8 T cell regulators. However, I would like to know a little bit more about why these 21 genes were chosen? There is one sentence, but this could be discussed a little bit more.

We have expanded the text to include further rationale for how we chose the 21 genes.

-Could the authors also state in more detail how the statistics were calculated for the animal experiments? Is it based on recipients or based on bonemarrow donors.

We have revised the figure legends to clarify this point. For the experiments in Figures 1 and 2 the statistics were calculated using reconstituted recipients as our n. For co-transfer competitive assays and the screen (Figures 3-5), statistics were calculated using recipients of the transferred T cells as our n.